# Effect of Graphite on the Recovery of Valuable Metals from Spent Li-Ion Batteries in Baths of Hot Metal and Steel

**Elsayed Mousa** [1,2,*] **, Xianfeng Hu** [1] **and Guozhu Ye** [1]

1   SWERIM AB, Aronstorpsvägen 1, SE-974 37 Luleå, Sweden; xianfeng.hu@swerim.se (X.H.);
    guozhu.ye@swerim.se (G.Y.)
2   Central Metallurgical Research and Development Institute (CMRDI), Cairo 12422, Egypt
*   Correspondence: elsayed.mousa@swerim.se

**Abstract:** The recycling of valuable metals from spent lithium-ion batteries (LIBs) is highly important to secure the sustainable production of new LIBs and reduce the dependence on virgin resources. The present paper aims to study the smelting behaviour of black mass (BM) from spent LIBs and investigate the effect of graphite on metal recovery in a carbon-saturated hot metal bath and in a low-carbon steel bath. The smelting trials of BM were conducted in a technical scale (150 kg) induction furnace using hot metal and steel scrap at operating temperatures in the range of 1278–1438 °C and 1470–1610 °C, respectively. Two grades of BM were applied in the current study; high-Ni BM and high-Co BM. Parts of both grades of the BM were briquettes to enhance the direct reduction of metal oxides with embedded graphite and to reduce the dust generation during loading into the furnace. The briquette BM was charged to carbon-saturated hot metal bath while the other part of the BM was subjected to de-coking in a muffle furnace in an oxidising atmosphere to remove graphite (37–39%) and to concentrate the valuable metals in the BM. The de-coked BM was loaded directly, without the need for the briquette, to the low-carbon steel bath. The results indicated that smelting of the de-coked BM in a steel bath is more efficient in metal recovery than the smelting of the corresponding briquette BM in a molten hot metal bath. The highest recovery rate of Co, Ni and Cu (98–99%) was obtained by smelting de-coked high-Co BM in a low-carbon molten steel bath, while the lowest recovery rate (38–55%) was obtained by smelting the briquette high-Ni BM in the carbon-saturated hot metal bath.

**Keywords:** LIBs; black mass; graphite; recycling; reduction; alloying of steel; pyrometallurgy

## 1. Introduction

Lithium-ion batteries (LIBs) are one of the most promising energy sources for various applications due to their high operating voltage, high specific energy, high power density and long cycle life [1,2]. Accordingly, the market value of global LIBs is sharply increasing [3] and consequently, the need for solutions dealing with end-of-life LIBs is crucially required [4]. The industry analysts predict that by 2030, the annual worldwide generated spent LIBs will be over 2 million tons [5] and it could even reach about 4 million tons [6]. The improper treatment of LIBs, either by landfill or incineration, can lead to serious risks to the environment and human health due to the presence of heavy metals and toxic electrolytes [7]. Meanwhile, the recycling of LIB components results not only in relieving the environmental contamination pressure but also in significant economic and social benefits. The spent LIBs usually contain 5–20% Co, 5–10% Ni, 5–7% Li, 5–10% other metals (e.g., Cu, Al, Fe, etc.), ~15% organic compounds and ~7% of plastic materials [8]. The recovery of valuable components from the spent LIBs can significantly contribute towards: (i) a supply of raw material, particularly the critical elements, which are needed for sustainable new battery production or other highly valuable applications, (ii) environmental concerns as the improper disposal of waste battery materials can result in the release of toxic gases,

corrosive liquids and dissolved metals that are poisonous to humans, plants, and animals, and (iii) economic profits and environmental gains as the recovery of valuable elements from secondary resources is more economic and generates fewer greenhouse emissions than its extraction from virgin resources. Therefore, from the resource, economic and environmental perspectives, it becomes urgent to find flexible and more efficient recycling routes for spent LIBs and for finding optimal methods and feasible techniques to recover the valuable components [9]. From a geopolitical standpoint and according to the European Commission, cobalt and natural graphite (both are main components in the LIBs cell) are critical elements while lithium and copper are expected to suffer from supply risk in the near future [10]. Other components in LIBs, such as Al, Ni and Mn, are economically important. At least one-third of the cost of the LIBs is related to the material cost [11]. Therefore, recovery of these valuable metals can release the heavy burden of industrial raw material shortage, save energy, reduce waste generation, decrease environmental risks, and maintain the circular economic approach [12,13]. All these factors make the recycling of LIBs essential to secure the sustainable production of LIBs, boosting the circular potential of batteries, conserving natural resources and reducing the environmental impacts.

More than 50 companies around the world are working on the recycling of spent LIBs on scales varying from small laboratory plants to industrial full-scale production [14]. The present global recycling rate of spent LIBs is estimated to be only 5–7% [6,15], which is far from the required recycling capacity that is significantly required for sustainable LIB production. Besides the low collection rate and safety concerns [14], the recycling of LIBs is challenged by other issues such as the complex structure of batteries, variety of battery types, shapes, sizes, and chemistries. The LIBs consist of compacted cells and each cell contains a cathode, anode, separator, electrolyte, and binder in a complex structure, which further complicates the recycling efforts of spent LIBs in order to recover all of the valuable components at a high yield and quality, meanwhile achieving profits compared to those of virgin minerals. Currently, there are three main recycling approaches for the recycling of spent LIBs: pyrometallurgy, hydrometallurgy, and direct recycling [16]. Pyrometallurgy applies high temperature to smelt the valuable metals in LIBs and form alloys [17,18] while hydrometallurgy employs multiple steps of acid-based leaching for individual metal separations [19,20]. On the other hand, direct recycling applies physical treatment processes with minimal destruction that enable the recovery of valuable metals without causing chemical changes [21,22]. Some recycling processes combine pyro- and hydrometallurgical steps and are often integrated with pre-processing and mechanical treatment [12,23]. The pre-processing includes steps of discharging, sorting, and classifications that do not alter the battery cells structure while mechanical, thermal, and physical processing involves dismantling, crushing, sieving, magnetic separation, flotation and others, which are used to liberate, classify, and concentrate the materials without altering their chemistry [24]. The pre-processing of spent LIBs generates fractions called "black mass, BM", which is composed of ~40–50% graphite, ~45% active materials, NMC, in the form of $Li(Ni_x, Mn_y, Co_{(1-x-y)})O_2$, and traces of Cu, Al, F, etc. [25–27].

In the pyrometallurgical process, the black mass is subjected to smelting reduction at high temperatures to recover Co, Ni, and Mn in the form of a metal alloy [28,29]. The recovered alloy can be used as a master alloy for producing special steels or it can be further processed to hydrometallurgy treatment to extract Co, Ni, and Mn in a pure state for new LIB production. Although the recycling of spent LIBs by pyrometallurgical routes is characterised by flexibility for different types and sizes of batteries and is able to achieve profits from the recovery of Co, Ni, and Cu at a large industrial scale, it suffers from the loss of Li and Al and partially Mn to the slag and the burning of graphite [30,31]. Moreover, only part of the graphite is used in the reduction of metal oxides, meanwhile, the rest is burnt and generates $CO_2$ or is entrapped in the slag. Recent newly developed concepts, based on a pyrometallurgical approach, called InduRed [32] and InduMelt [33] showed promising results in the recovery of Li. Most of Li is recovered in the gas phase and collected in the top dust while less than 10% is lost in the slag. In the ReLion pyrometallurgical process

developed at SWERIM [34,35], it was possible to recover >95% of Co, Ni, and Mn in metal alloys and ~70% of Li, recovered in the flue dust as $Li_2CO_3$, but still, graphite has not been recovered. In hydrometallurgical processes, the black mass could be subjected to multiple steps of acid leaching, solvent extraction, and precipitation to extract Co, Ni, and Mn from the black mass as corresponding sulfates, which can be further purified and used as the precursors for manufacturing of new LIBs [27,28]. However, the presence of high-volume graphite in the black mass restricts the efficiency of the leaching process and makes the process more costly and time-consuming. Although hydrometallurgy is characterised by lower energy consumption compared to pyrometallurgy, the long chain of metals recovery, wastewater generation, and acid consumption are the main drawbacks of hydrometallurgy for spent LIB recycling. The direct recycling process is another promising route to recover the useful components from spent LIBs without high energy consumption or chemicals unlike pyro- and hydrometallurgical routes, respectively, however, the performance of the recycled materials could be affected by the presence of impurities in the long term. In addition, direct recycling has not been demonstrated at a large scale as it required single-cathode input and good control of battery quality [16,36].

In this context, the pyrometallurgical processes can perform the simplest, profitable, and highest production routes among other routes on an industrial scale, however, the excess graphite is not only burnt and generates $CO_2$ emissions but also generates unnecessary extra heat in the reactor and the off-gas system. Additionally, it could affect the full recovery of valuable metals from the black mass (BM) [31]. Therefore, this paper aims at understanding the smelting behaviour of the BM and the effect of graphite on metal recovery by melting the BM in a carbon-saturated molten bath of hot metal (pig iron) or in a low-carbon molten bath of steel scrap. Currently, there are two mainstream LIBs in the market for almost all EVs; lithium iron phosphate (LFP) batteries and NMC lithium batteries. The paper demonstrates knowledge on the smelting of NMC from spent LIBs and the behaviour of graphite and evaluates the metal recovery and the slag formation in molten iron and steel baths. The trials were executed on a technical scale in view of the results obtained at the lab scale [34], meanwhile, the results of the current work contributed to the designing and setting of the operating parameters of the pilot-scale trials in EAF, as discussed elsewhere [35].

## 2. Materials and Methods

### 2.1. Materials Characterisation

Two different grades of BM were sourced from a recycling facility where the spent LIBs are treated by mechanical processing. The first grade is a type of NMC532 while the second grade is a mixture of different spent LIB types with high Co content. The mechanical processing involves crushing, sieving and magnetic separation where the outputs from the separation stages consist of different metal concentrates. The BM from sieving is the undersize (−4.0 mm) fractions in which cathode active materials are concentrated with graphite. The two grades of BM were subjected to chemical analysis using the LECO (CS230 Series) combustion method (only for carbon and sulfur analysis) and the ICP-SFMS method, as given in Table 1. The first grade was relatively higher in Ni (namely high-Ni BM) while the second grade was higher in Co (namely high-Co BM). Graphite was the main component (37–39%) in both grades of the BM. The X-ray powder Diffraction (Analytical Empyrean XRD) method is used for phase identification, as shown in Figure 1. The high-Ni BM consists mainly of Li–Ni oxide and Li–Mn–Ni oxide phases while the high-Co BM consisted of Li–Co oxide and Li–Al–Co oxide. Graphite was the main phase in both grades of BM. Hot metal and steel scrap, with the chemical composition given in Table 2, were used as carbon-saturated and low-carbon melting baths of the BM, respectively.

**Table 1.** Chemical composition of high-Ni and high-Co BM.

| Element, wt.% | Co | Cu | Li | Ni | Mn | Al | Fe | P | S | F | C | $O_2$ |
|---|---|---|---|---|---|---|---|---|---|---|---|---|
| High-Ni BM | 3.64 | 4.82 | 2.53 | 9.97 | 6.94 | 2.26 | 1.12 | 0.40 | 0.39 | 1.76 | 39.28 | 26.89 |
| High-Co BM | 19.71 | 2.23 | 3.17 | 2.88 | 1.88 | 1.32 | 0.45 | 0.37 | 0.22 | 0.88 | 37.03 | 29.86 |

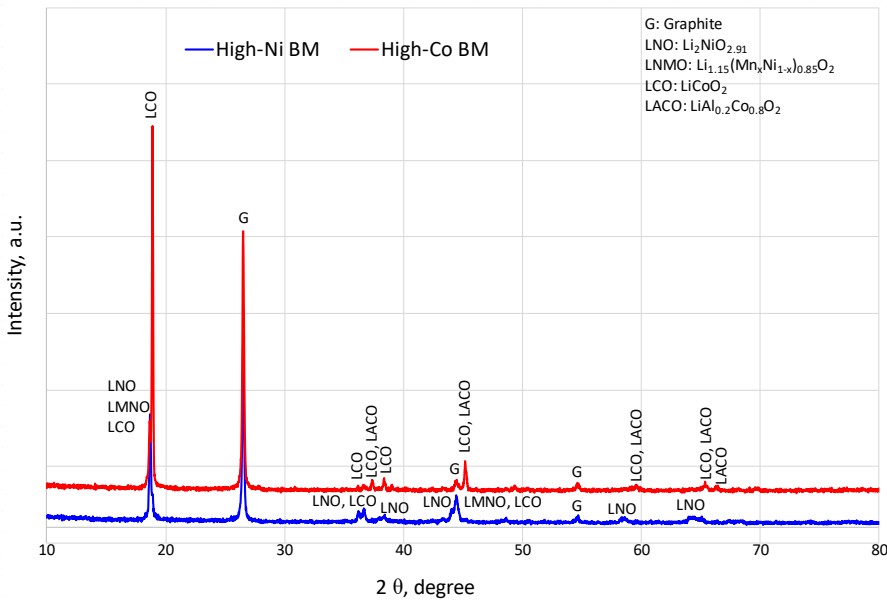

**Figure 1.** XRD analysis results of high-Ni and high-Co BM.

**Table 2.** Chemical compositions of hot metal (pig iron) and steel scrap.

| Element, wt.% | Fe | C | Si | S | P | Mn | Cr | Co | Ni | Ti | Cu |
|---|---|---|---|---|---|---|---|---|---|---|---|
| Hot metal (pig iron) | 95.67 | 4.11 | 0.16 | 0.01 | 0.03 | 0.02 | 0.04 | - | - | 0.02 | - |
| Steel scrap | 98.25 | 0.05 | 0.03 | 0.00 | 0.01 | 1.28 | 0.06 | 0.02 | 0.05 | 0.03 | 0.03 |

## *2.2. Experimental Procedure*

### 2.2.1. Briquettes of Black Mass

To reduce the dust generation during loading of the BM, briquettes were produced from both grades of BM by a hydraulic press (Stenhøj press, 60 T) and a using stainless-steel mould (Ø = 7 cm). The briquettes were created successfully without the addition of any binder to the black mass. The average density of the briquettes was ~1.8 g/cm$^3$ and ~2.4 g/cm$^3$ for high-Ni BM and high-Co, respectively. The pre-calculated amount of briquette BM-included graphite was loaded with the hot metal pieces while the BM fines are subjected to de-coking to remove the graphite before loading with the steel scrap.

### 2.2.2. De-Coking of Black Mass

To keep the carbon content at a low level in steel, and to see the impact of graphite on the recovery rate of metals from the BM, the high-Ni BM and high-Co BM were subjected to mild oxidation with air to remove the graphite. The pre-treatment to remove the graphite from the BM is called "de-coking". The de-coking was performed in a chamber furnace at 800 °C under a flow of air for 12 h. The carbon content in the BM after de-coking was measured by LECO, which indicated the full combustion of all graphite (C ≈ 0%) from the BM after de-coking. The individual and cumulative size distribution of de-coked materials are given in Figures 2 and 3, respectively. About 90% of both grades (high-Ni BM and high Co BM) of de-coked BM materials have sizes under 1.0 mm.

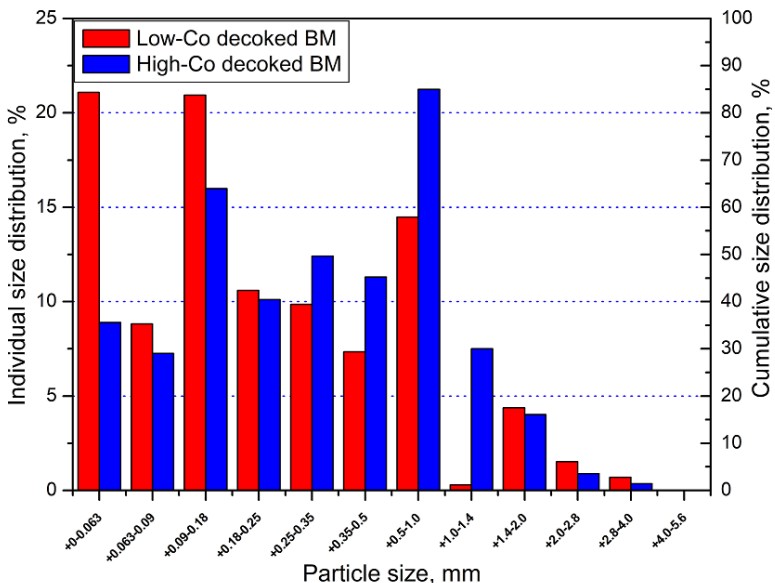

**Figure 2.** Individual size distribution of high–Ni and high–Co BM after de–coking.

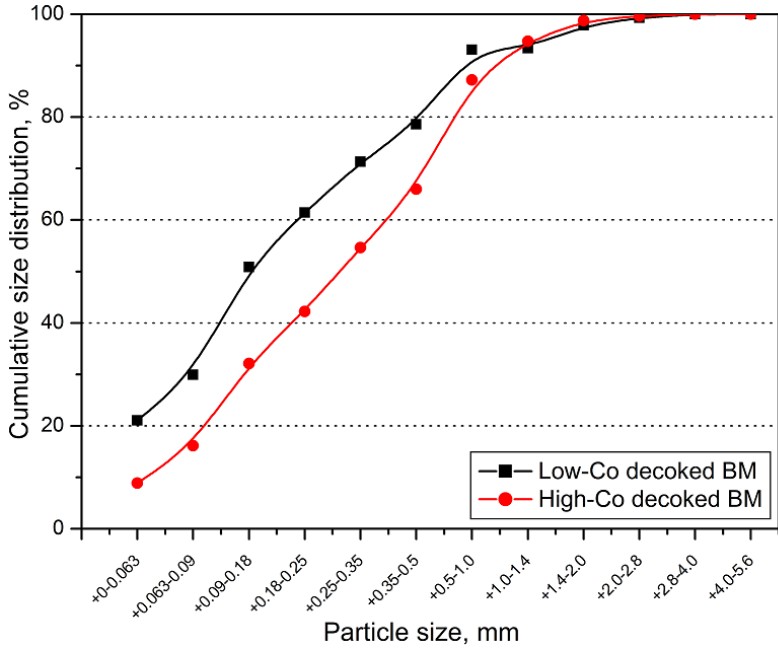

**Figure 3.** Cumulative size distribution of high–Ni and high-Co BM after de–coking.

### 2.2.3. Design of Smelting Trials

To investigate the smelting behaviour of high-Ni and high-Co BM and the effect of graphite on the metal recovery in baths of hot metal and steel; four trials were designed as shown in Table 3. In trials 1–2, the briquette BM (high-Ni and high-Co BM with graphite) was loaded to a bath of carbon-saturated hot metal to achieve self-reduction of the metal oxides and to investigate the behaviour of the excess graphite in the BM when it came into contact with the carbon-saturated bath. On the other hand, trials 3–4 were designed to investigate the smelting behaviour of de-coked BM (without graphite) in a molten bath of low-carbon steel scrap. To achieve the reduction of the embedded metal oxides in the BM, commercial-grade anthracite (95% C-fix) was added and well mixed with the de-coked BM materials. Based on the chemical composition of the de-coked BM and by assuming that Ni, Mn, Co, Li and Cu are the main reducible oxides in the de-coked BM, the anthracite was stoichiometrically calculated and multiplied by 1.5 to ensure the complete reduction

of $Co_2O_3$, $NiO$, $Li_2O$ and $Cu_2O$ during the smelting reduction process. By comparing trial 1 vs. 2 and trial 3 vs. 4, it would be possible to evaluate the impact of the BM grade (high-Ni and high-Co BM) on metal recovery and the behaviour of graphite when it came into contact with the carbon-saturated hot metal bath and the low-carbon molten steel bath, respectively. By comparing trial 1 vs. 3 and trial 2 vs. 4, it would be possible to evaluate the impact of graphite on the metal recovery from the same grade of BM. A 150-kg technical scale induction furnace (ASEA model, melting power 90 kW, and frequency 2220 Hz) was used to perform the designed trials. The furnace was placed on four mechanical jacks for accurate vertical positioning and the furnace could be tilted by hydraulic cylinders to control the tapping of the molten alloy. The furnace was lined with MgO-type refractory bricks and $N_2$ was used as shield gas to avoid the re-oxidation of the melt. A schematic diagram and a full description of the furnace are given elsewhere [37].

**Table 3.** Trials designed for smelting of BM in induction furnace.

| Trial No. | Type of BM | Melting Bath | Reducing Agent |
|---|---|---|---|
| 1 | Briquettes high-Ni BM | Hot metal | Embedded graphite in BM |
| 2 | Briquettes high-Co BM | Hot metal | Embedded graphite in BM |
| 3 | Fines de-coked high-Ni BM | Steel scrap | Anthracite |
| 4 | Fines de-coked high-Co BM | Steel scrap | Anthracite |

Figure 4a schematically shows the design of trials 1 and 2 conducted with briquette BM in a hot metal bath while Figure 4b shows the design of trials 3 and 4 conducted with de-coked BM in a steel scrap bath.

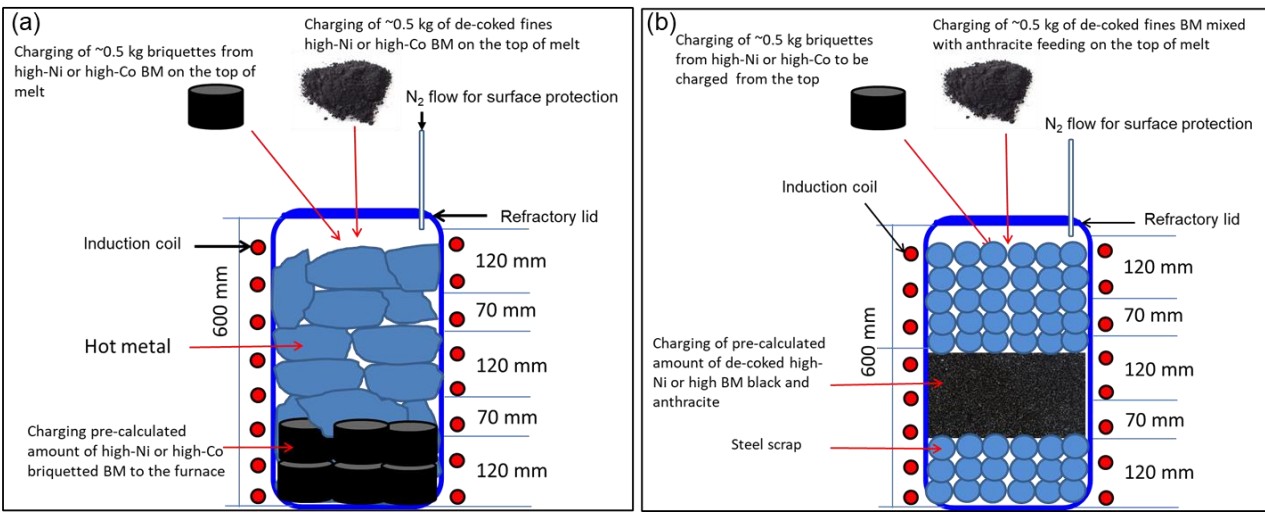

**Figure 4.** Schematic diagram for: (**a**) trials 1 and 2 (**b**) trials 3 and 4.

The experimental procedure of smelting can be described as follow:

(i) Weighing ~70 kg of hot metal and ~4 kg of briquette high-Ni BM in trial 1 or high-Co BM in trial 2. The amount of hot metal was calculated to reach the required filling level in the furnace that enabled an efficient heating rate.

(ii) Loading the briquette high-Ni BM in trial 1 or high-Co BM in trial 2 into the furnace followed by hot metal pieces as schematically described in Figure 4. The BM was loaded at the bottom and covered by heavier hot metal pieces to restrict the floating of the briquettes to the top during the smelting reduction process.

(iii) Heating the furnace up to complete the melting of materials under a flow of $N_2$ to avoid the oxidation of the melt.

(iv) Measuring the temperature of the melt using a temperature CELOX probe.

(v) Sampling of the slag and metal for subsequent chemical analysis using a steel scoop.

(vi)　Removing the slag then adding 2 briquettes (total weight ~0.5 kg) of high-Ni BM in trial 1 or high-Co BM in trial 2 to investigate the melting behaviour of the BM on the top of the melt. The first slag collected from trials 1 and 2 was de-coked in a muffle furnace at 800 °C under a flow of air to remove the graphite. The de-coked slags were subjected to chemical analysis.

(vii)　Waiting 20 min to ensure the complete melting and measuring temperature of the melt and take samples from the slag and metal for subsequent analysis.

(viii)　Adding ~0.5 kg of de-coke high-Ni in trial 1 or high-Co in trial 2 BM to the melt to investigate the melting behaviour of the de-coked BM on top of the melt.

(ix)　Waiting for 20 min to ensure the complete melting of materials, measure the temperature of the melt and take a sample from the slag and metal for subsequent analysis.

(x)　Taping the melt and conducting mass balance for input and output materials.

Trials 3 and 4, using de-coked BM in steel scrap, were conducted using essentially the same procedures as that described for trials 1 and 2. However, 70 kg of steel scrap was used in trial 3 and 90 kg of steel scrap was used in trial 4. The 20 kg of extra scrap in trial 4 was added to reach a certain level in the furnace to enable sampling during melting. De-coked BM (4.5 kg) was well mixed with the anthracite (0.7 kg) and loaded in the middle of the steel scrap bed, as shown in Figure 5. The smelting process in the induction furnace starts with the melting of the BM with the scrap (Figure 5a), top loading of de-coke fines and/or briquettes BM on top of the melt (Figure 5b), complete melting of materials (Figure 5c) and final tapping of alloyed steel (Figure 5d).

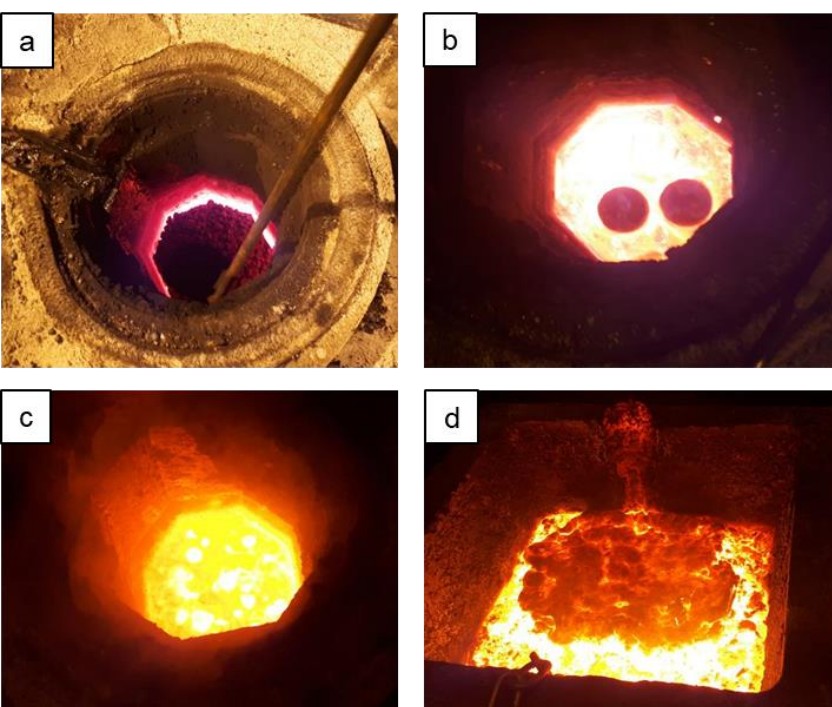

**Figure 5.** Photographs during smelting process in induction furnace: (**a**) loading of BM with scrap (**b**) loading of briquettes on top of melt (**c**) complete melting of scrap and BM (**d**) tapping of alloyed steel.

The metal and slag samples were chemically analysed by X-ray spectrometry (XRF, ASTM E 572), carbon and sulfur were analysed by LECO (CS230 analyser), chemical phases were identified by X-ray diffraction analysis (XRD Analytical Empyrean).

## 3. Results and Discussion

### 3.1. Melting Behaviour

The final metal samples obtained before taping and after the complete dissolving of the BM in the hot metal or molten steel scrap baths were subjected to chemical analysis as given in Table 4. The concentration of Co, Ni and Cu were increased by dissolving the BM in a hot metal and a scrap bath. The metal oxides in the BM are reduced by the graphite/anthracite as given in Equation (1). The carbon content in hot metal was not affected by the smelting of the BM due to the saturation of iron with the carbon in the hot metal. Meanwhile, in the molten steel bath, the carbon content was significantly increased due to the excess addition of anthracite (1.5 stoichiometric) which was added to ensure the complete reduction of metals oxides in the de-coked BM. The concentration of Mn slightly increased by melting the BM in hot metal while it decreased in the case of melting BM in the steel scrap, which can be explained by the effect of dissolved carbon in the molten iron and steel baths. The slight increase in Mn in the hot metal was attributed to the reverse reaction in Equation (2). The molten iron saturated with carbon, in addition to the extra graphite from the BM, enabled the partial reduction of MnO from the generated slag to the Mn metal, however, most of Mn turned to MnO in the slag phase. In the molten steel bath, the Mn concentration was decreased. This can be attributed to the partial reoxidation of Mn metal that either presents in the starting steel scrap or comes from the smelting reduction of the BM. The deficiency of carbon in the steel bath could facilitate the reoxidation of Mn to MnO and consequently decrease the Mn concentration in the final steel product.

$$M_xO_y + yC = xM + yCO \tag{1}$$

where, $M$ can be Co, Ni, Cu, Mn, Li, and Fe

$$[Mn] + \{CO\} = (MnO) + [C] \tag{2}$$

**Table 4.** Analysis of metal samples from trials 1–4 in comparison to hot metal and steel scrap.

| Component/Trial | Co | Ni | Cu | Mn | Si | C | S | Fe | Others |
|---|---|---|---|---|---|---|---|---|---|
| | | | | | wt.% | | | | |
| Hot metal (Ref.) | 0 | 0 | 0 | 0.02 | 0.16 | 4.11 | 0.01 | 95.67 | 0.03 |
| Melt with high-Ni BM (trial 1) | 0.13 | 0.30 | 0.21 | 0.22 | 0.08 | 4.07 | 0.01 | 94.81 | 0.17 |
| Melt with high-Co BM (trial 2) | 0.95 | 0.16 | 0.12 | 0.21 | 0.02 | 4.11 | 0.01 | 94.25 | 0.17 |
| Steel scrap (Ref.) | 0.01 | 0.05 | 0.03 | 1.28 | 0.03 | 0.05 | 0.00 | 98.25 | 0.29 |
| Melt with high-Ni BM (trial 3) | 0.45 | 1.02 | 0.45 | 0.44 | 0.01 | 0.53 | 0.01 | 96.86 | 0.23 |
| Melt with high-Co BM (trial 4) | 1.96 | 0.29 | 0.25 | 1.09 | 0.02 | 0.72 | 0.01 | 95.46 | 0.20 |

The loading of de-coked BM fines on the top of the melt showed complete dissolving of the de-coked BM in the hot metal and steel baths. On the other hand, the briquette BM was floated on the surface of the melt when it was loaded on the top of the melt, as shown in Figure 5b. In the case of the hot metal (trials 1 and 2), the floated briquettes were partially dissolved in the hot metal bath due to the relatively lower operating temperature (<1450 °C) and the rest of the briquettes were removed with the slag. In the steel melt (trials 3 and 4), the floated briquettes were completely dissolved in the steel melt by raising the temperature to >1500 °C. This indicates that graphite can be completely dissolved in the low-carbon steel molten bath enabling the reduction of metal oxides and almost the full recovery of metals. In view of these results, injection of BM fines through submerged injection lance in the pilot EAF was recommended and the operating temperature was controlled at >1500 °C, as discussed elsewhere [33]. Figure 6 shows the operating temperature of trials 1–4. The operating temperature to reach the melting status in a hot metal bath was lower than that required for a steel bath, which was attributed to the effect of the carbon content in the starting hot metal and steel scrap. Both operating temperature and graphite content in the

BM affected the recovery rate of metals in the hot metal and steel baths, as will be discussed in the next section of metal recovery.

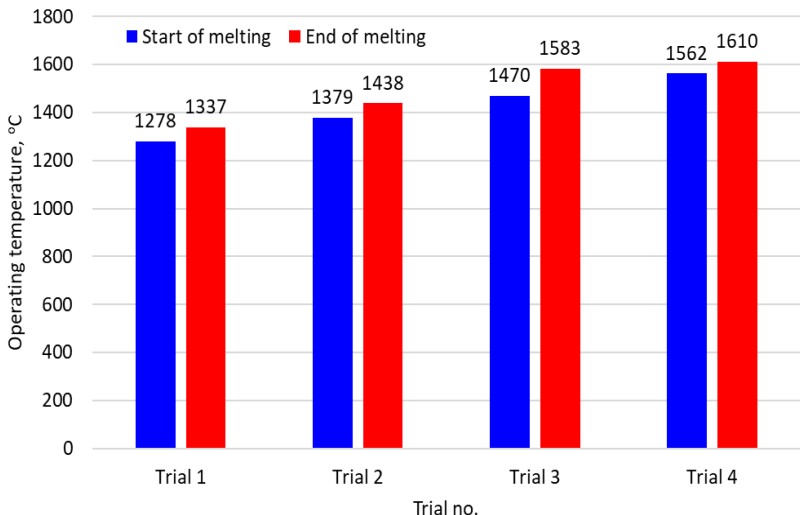

**Figure 6.** Operating temperature during smelting process in induction furnace.

### 3.2. Metal Recovery from BM

The approximated mass balance of input and output materials excluding gases that were emitted during the smelting process is given in Table 5. The hot metal used in trials 1 and 2 was melted at a lower temperature compared to the steel scrap used in trials 3 and 4 due to the effect of the carbon content. This significantly affected not only the recovery rate of metals from the BM but also the amount of slag generated. The generated slag from melting of the briquette BM in the hot metal bath was higher than that produced with the steel bath. In addition, the excess graphite (37–39% in loaded briquette BM) was floated on the top of the melt carrying away part of metal oxides to the slag phase. Therefore, the excess graphite and the relatively lower operating temperature had restricted the complete reduction and melting of the BM metal oxides and consequently reduced the recovery rate of metals. On the other hand, de-coking of the BM and using low-carbon steel scrap enhanced the recovery rate of metals in the alloyed steel product and low slag was generated from the melt.

**Table 5.** Mass balance of input and output materials in trials 1–4.

| Trial No. | Operating Temperature, °C | Input Materials, kg | | | | | Output Products, kg | |
|---|---|---|---|---|---|---|---|---|
| | | Pig Iron | Steel Scrap | BM-Briq | BM-De-Coked | Anthracite | Alloyed Metal | Slag |
| 1 | 1278–1337 | 70.2 | 0.0 | 4.5 | 0.5 | 0.0 | 66.4 | 4.0 |
| 2 | 1379–1438 | 70.3 | 0.0 | 4.5 | 0.5 | 0.0 | 68.9 | 3.3 |
| 3 | 1470–1583 | 0.0 | 70 | 0.5 | 4.0 | 0.6 | 57.9 | 2.9 |
| 4 | 1562–1610 | 0.0 | 90 | 0.5 | 4.5 | 1.1 | 89.8 | 0.7 |

The carbon and sulfur were analysed in the first generated slag samples from trials 1–4, as given in Table 6. The carbon content of the slag samples from trials 1 and 2 was high due to the presence of graphite, while it is low in slag generated from trials 3 and 4 due to the use of the de-coked BM. The first slag generated from trials 1 and 2 was subjected to de-coking in a muffle furnace to remove the attached graphite. The first slag samples, which were collected from trials 1 and 2, were analysed by XRD before and after de-coking, as shown in Figures 7 and 8, respectively. After smelting of the BM in a hot metal bath in trials 1 and

2, graphite became the main phase in the slag, while it was completely combusted after the de-coking of the slag. The XRD of de-coked slags exhibited the presence of lithium aluminium silicate ($LiAlSiO_4$) and lithium aluminium oxide ($LiAlO_2$) besides others phases such as lithium cobalt manganese oxide ($Li_2CoMnO_3$) and lithium nickel oxide ($LiNiO_2$) in both grades of BM. This indicated that part of the valuable metals (Li, Co, Ni, Mn) from the briquette BM was carried away by the graphite to the top of the melt and collected in the slag. The formation of $LiAlSiO_4$ and $LiAlO_2$, which is a complex solid solution phase of lithium aluminum silicate and oxides, was attributed to the presence of alumina and silica, which fixed the Li in the slag. This is in line with the findings reported by Vest el al. [38], who indicated the high affinity of Li to alumina and silica in slag. It was not possible to capture dust during the smelting trials as the off-gases went through the central suction system but our ongoing research is focused on enhancing the Li evaporation in the gas phase by slag design and collecting the Li dust from the flue gas.

**Table 6.** Carbon and sulfur analysis in first slag samples.

| Trial No. | C, % | S, % |
|---|---|---|
| 1 | 27.50 | 0.06 |
| 2 | 22.20 | 0.04 |
| 3 | 0.07 | 0.03 |
| 4 | 0.05 | 0.02 |

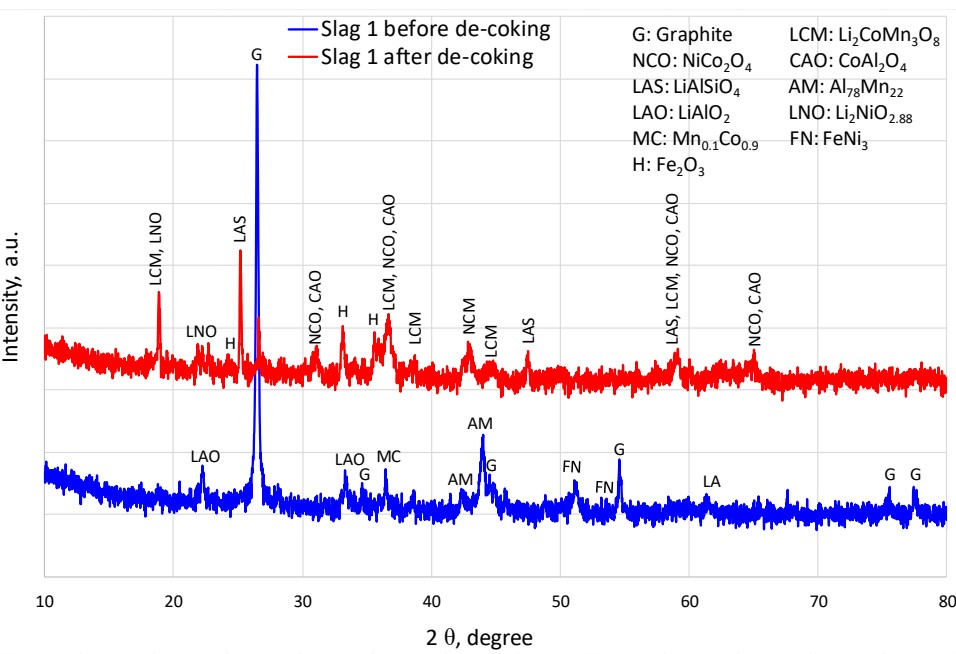

**Figure 7.** XRD analysis of the first slag before and after de-coking of trial 1 with high-Ni BM.

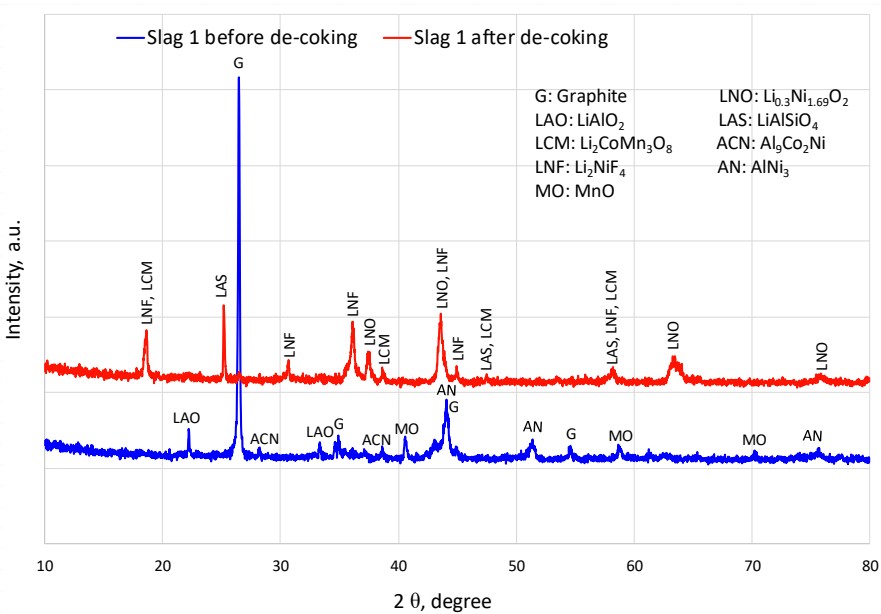

**Figure 8.** XRD analysis of the first slag before and after de-coking of trial 2 with high-Co BM.

The first slag generated from trials 3 and 4 is chemically analysed as given in Table 7. The main composition of the first slag collected from trial 3 was composed of $Al_2O_3$, MnO and $SiO_2$ while the first slag collected from trial 4 consisted mainly of FeO, $Al_2O_3$, $SiO_2$ and MgO. The last slag samples collected from trials 1–3 before tapping were chemically analysed, as given in Table 8, while no slag was generated at the end of trial 4. The main composition of all slags was $SiO_2$, $Al_2O_3$, FeO and MnO besides MgO in slag 1. The concentration of Co and Ni oxides in the slag of trial 1 (high-Ni BM) was remarkable compared to that in trial 2 (high-Co BM).

**Table 7.** Chemical analysis of first slag generated from trials 3 and 4.

| Sample | C | S | CaO | MgO | $SiO_2$ | $Al_2O_3$ | FeO | MnO | $Cr_2O_3$ | $V_2O_5$ | $Co_2O_3$ | NiO | $Cu_2O$ |
|---|---|---|---|---|---|---|---|---|---|---|---|---|---|
| | | | | | | | wt.% | | | | | | |
| Trial 3, slag 1 | 0.07 | 0.03 | 4.30 | 1.43 | 19.60 | 33.40 | 4.43 | 24.70 | 0.21 | 0.61 | 0.01 | 0.03 | 0.01 |
| Trial 4, slag 1 | 0.05 | 0.02 | 0.81 | 7.35 | 8.92 | 12.50 | 62.10 | 4.27 | 0.45 | 1.67 | 0.33 | 0.12 | 0.09 |

**Table 8.** Chemical analysis of last slag (slag 3) generated from trials 1–3.

| Sample | C | S | CaO | MgO | $SiO_2$ | $Al_2O_3$ | FeO | MnO | $Cr_2O_3$ | $V_2O_5$ | $Co_2O_3$ | NiO | $Cu_2O$ |
|---|---|---|---|---|---|---|---|---|---|---|---|---|---|
| | | | | | | | wt.% | | | | | | |
| Trial 1, slag 3 | 0.28 | 0.04 | 5.35 | 9.30 | 26.00 | 20.40 | 14.30 | 13.00 | 3.63 | 0.62 | 0.39 | 0.95 | 0.17 |
| Trial 2, slag 3 | 0.39 | 0.02 | 4.46 | 1.01 | 28.20 | 25.90 | 26.60 | 11.50 | 0.30 | 0.78 | 0.09 | 0.03 | 0.01 |
| Trial 3, slag 3 | 0.06 | 0.03 | 4.00 | 1.13 | 25.00 | 27.20 | 9.69 | 26.40 | 0.56 | 1.09 | 0.01 | 0.05 | 0.02 |

In trials 3 and 4, the steel scrap was melted at a higher temperature compared to that of the hot metal, as shown in Figure 6, due to lower carbon content. The higher temperature enhanced the dissolving of the de-coked BM in the molten steel bath. In trial 3, it was difficult to raise the temperature up to 1600 °C due to the limited amount of loaded scrap (~70 kg), which made the process of metal sampling very difficult. In trial 4, more steel scrap was loaded (~90 kg), which gave the opportunity to increase the operating temperature up to 1600 °C and achieve a complete melting of de-coked BM and promote the sampling process.

The recovery rate of Co, Ni and Cu in the alloyed product of iron and steel is calculated according to Equation (3) and the results are given in Table 9. The highest recovery rate of Co, Ni, and Cu was 99%, 98% and 99%, respectively. This was achieved in trial 4 when de-coked high-Co BM was smelted in a bath of steel scrap. By smelting the de-coked high-Ni BM in a steel bath (trial 3), the recovery rate of Co was slightly reduced while Ni and Cu were significantly reduced to 81% and 72%, respectively, which can be attributed to the relatively lower operating temperature (1470–1583 °C) compared to trial 4 (1562–1610 °C). The lowest recovery rate of Co, Ni, and Cu was 45%, 38% and 55%, respectively, in trial 1 which was designed by smelting the briquette BM in a hot metal bath. This can be attributed to the lower operating temperature (1278–1337 °C) and non-complete reduction and melting of the BM with excess graphite. The excess graphite was floated on the upper surface of the hot metal carrying away part of the metal oxides to the slag phase without accomplishing complete reduction.

**Table 9.** Metal recovery from BM in hot metal and scrap.

| Type of BM | High-Ni BM | | High-Co BM | |
|---|---|---|---|---|
| Melting Bath | Hot Metal | Steel Scrap | Hot Metal | Steel Scrap |
| Trial No. | 1 | 3 | 2 | 4 |
| Element | Metals Recovery, % | | | |
| Co | 45.0 | 99.2 | 62.3 | 99.9 |
| Ni | 37.8 | 80.3 | 71.7 | 98.4 |
| Cu | 54.8 | 71.3 | 69.5 | 99.9 |

In general, the metal recovery in the case of using de-coked BM in a steel scrap bath (trials 3 and 4) was higher than that in the case of using briquette BM in a hot metal bath (trials 1 and 2). Moreover, the recovery rate of metals was higher in the case of using high-Co BM (trials 2 and 4) compared to that of high-Ni BM (trials 1 and 3). It can be concluded that the lower operating temperature and excess graphite have a negative impact on the recovery rate of metal from the BM. The highest recovery of Co, Ni, and Cu (98–99%) was obtained at an operating temperature in the range of 1562–1610 °C, while the lowest recovery rate (38–55%) was obtained at an operating temperature in the range of 1278–1337 °C. Besides the effect of operating temperature, the removal of graphite from the BM by de-coking enabled the BM to stay in the melt and not float on the top of the melt during the smelting process. Consequently, the recovery rate of metals from the de-coked BM in the steel bath was higher than that with the briquette BM in the hot metal bath. In view of these findings, injection could be an efficient way to reach a higher metal recovery, meanwhile, it can enable the separation of excess graphite on top of the molten alloy.

$$Recovery\ rate\ (\%) = \frac{(C_a * w_a) - (C_m * w_m)}{(C_b * w_b) - ((C_d * w_d)} \times 100 \tag{3}$$

where $C_a$, $C_m$, $C_b$, and $C_d$ is the concentration of the recovered metal (*Co, Ni, Cu*) in the alloyed product, in the starting metal, in the briquette BM, and in the de-coked BM, respectively; $w_a$, $w_m$, $w_b$, and $w_d$ is the weight of the alloyed product, the starting metal, the briquette BM, and the de-coked BM, respectively.

By evaluating the recovery rate of metals from the BM, it can be generally concluded that: (i) the recovery rate was higher with high-Co BM than that with high-Ni BM regardless of the type of melting bath, (ii) de-coked BM can improve the recovery rate of valuable metals compared to the BM with graphite, (iii) graphite from the BM can be dissolved completely in a low-carbon steel melt, (iv) in a carbon-saturated melt, the excess graphite from the BM floats on top of the molten alloy during the smelting process and can be separated during the smelting reduction under an inert atmosphere, however, it takes part of the metal oxides to the slag phase and consequently it negatively affects the recovery rate of metals, (v) the operating temperature has a significant effect on the recovery rate

of metals from the BM and it is recommended to work at temperatures in the range of 1500–1600 °C, and (vi) it is recommended to separate graphite from the BM [39] before the smelting process to achieve high recovery rate of metals, however, the injection of the BM containing graphite into a low-carbon molten bath could be also an efficient scenario to reach a high recovery of metal by enhancing the reduction kinetics of metal oxides with the carbon, meanwhile, the surplus graphite can be separated as a solid powder with the slag from the top of the molten alloy.

The ongoing research aims to enhance the Li evaporation and collection in the dust by slag design using the FactSage programme for thermodynamic calculations. In addition, froth flotation will be used for the separation of graphite from the pre-treated BM instead of the de-coking process.

## 4. Conclusions

Developing a smelting reduction process for efficient recycling of spent Li-ion batteries (LIBs) and the recovery of valuable metals such as Co, Ni, Mn, Cu and graphite from LIBs is crucial for saving resources, reducing the environmental impact and securing sustainable production of LIBs. This study discussed the melting behaviour of black mass in a carbon-saturated hot metal bath and a low-carbon steel scrap bath for the production of iron and steel-base alloys. The trials were conducted in a pilot-scale (150 kg) induction furnace. Four trials were designed for the recovery of valuable metals from high-Ni and high-Co black mass in alloying hot metal and steel. The graphite content in the BM as-received was in the range of 37–39%. Part of the black mass was briquette and was loaded in a hot metal bath while another part was subjected to de-coking in an oxidised atmosphere to remove the graphite and was loaded with steel scrap without briquettes. The main findings can be summarised in the following points:

1.  The highest recovery rate of Co, Ni and Cu was in the range of 98–99% by using de-coked high-Co black mass in a bath of molten steel scrap, while the lowest recovery rate in the range of 38–55% was obtained by smelting of briquette high-Ni black mass in the hot metal bath.
2.  The recovery rate of metals was higher in the case of the high-Co black mass compared to the high-Ni black mass, regardless of the type of molten bath.
3.  In the case of using the briquette BM with hot metal, the excess graphite floated on the upper surface of the hot metal, and it restricted the complete reduction of the metal oxides and consequently decreased the recovery rate of the metal.
4.  The recovery rate increased by increasing the operating temperature (1500–1600 °C). Injection of the BM with graphite to a low-carbon molten bath and working at a high temperature (>1500 °C) could be an efficient scenario to partially dissolve the graphite and recover the metals, meanwhile, separating the surplus graphite on top of the melt, however, further trials are required to approve this concept.
5.  The analysis of the de-coked slag indicated the presence of lithium in the complex phases of Ni, Co, Mn oxides and aluminum silicate.
6.  The future work will focus on designing slag for enriching the Li recovery in dust and improving the metal recovery from the black mass, meanwhile, froth flotation will be used for the separation of graphite from the pre-treated black mass.

**Author Contributions:** E.M.: Methodology, Investigation, Formal analysis, Writing—original manuscript. X.H.: Writing—review and editing. G.Y.: Methodology, Investigation, Writing—review and editing, Project administration, Funding acquisition. All authors have read and agreed to the published version of the manuscript.

**Funding:** The authors acknowledge the financial support from the Swedish Energy Agency to the Re-Lion project (project no. 2016-006027). The paper writing and revision were carried out with the support of the ERA-MIN 2 program and the respective VINNOVA national financier to the NEXT-LIB project (project no. 2019-03473).

**Conflicts of Interest:** The authors declare that they have no know competing financial interest or personal relationship that could have appeared to influence the work reported in this paper.

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
