# Peer review of "Effect of Graphite on the Recovery of Valuable Metals from Spent Li-Ion Batteries in Baths of Hot Metal and Steel"

_recycling, doi:10.3390/recycling7010005_

Round 1
Reviewer 1 Report
The present manuscript describes an interesting procedure for recycling lithium-ion batteries. Different conditions were evaluated, and valuable metals as well as graphite were recover. The manuscript describes the process in detail, and the results are clearly explained.
Only some comments:
- Figure 1: which crystalline phases correspond to peaks between the range of 30º-40º? Please, indicate them.
- The individual and cumulative size distribution of de-coked materials which are given in Figure 2 could be exhibit in two different Figures for more clarity for the reader.
- "Figure 3a" and "Figure 3b" text are in black in the text.
- Idem for "Figure 4" "Eq. 1" "Eq. 2"... and forward. Please, homogenize the text format
- Figure 6. Identify all peaks in the XRD patterns.
- References should exhibit the same format
After these considerations, the manuscript can be consider for publication.
Author Response
The authors express their appreciation to the reviewer for the valuable comments and suggestions to improve the quality of our manuscript. Below we have provided point by point response to the reviewer comments.
Comments and Suggestions for Authors
- The present manuscript describes an interesting procedure for recycling lithium-ion batteries. Different conditions were evaluated, and valuable metals as well as graphite were recover. The manuscript describes the process in detail, and the results are clearly explained.
Author’s response: The authors highly appreciate the given feedback and thoughts by reviewer. The authors have considered all comments on the manuscript to make it in the best quality.
- Only some comments:
- Figure 1: which crystalline phases correspond to peaks between the range of 30º-40º? Please, indicate them.
Author’s response: Many thanks for this comment. The phases were checked carefully, and Figure 1 is updated with all phases in detail. The text of description is updated in whole manuscript to make it clear. The name of low-Co BM is changed to high-Ni BM to clearly distinguish between the two different grades of BM (high-Ni BM and high-Co BM) which have been used in this study.
- The individual and cumulative size distribution of de-coked materials which are given in Figure 2 could be exhibit in two different Figures for more clarity for the reader.
Author’s response: Figure 2 is updated and divided into 2 figures as recommended. The number of other following figures is updated according to this change.
- "Figure 3a" and "Figure 3b" text are in black in the text.
Author’s response: The format is corrected.
- Idem for "Figure 4" "Eq. 1" "Eq. 2"... and forward. Please, homogenize the text format
Author’s response: The format is correct
- Figure 7. Identify all peaks in the XRD patterns.
Author’s response: All phases have been identified as given in Figures 7 & 8. The text is modified as highlighted in the manuscript.
- References should exhibit the same format
Author’s response: References are updates to be in same format.
After these considerations, the manuscript can be considered for publication.
Author’s response: The authors thank the reviewer for his valuable comments and suggestions which highly improved the quality of the manuscript.
Best regards,
Elsayed Mousa
Corresponding author
Swerim AB
Metallurgy department
Luleå, Sweden
elsayed.mousa@swerim.se
Reviewer 2 Report
Dear authors!
The manuscript has interesting experimental results, but thermodynamic analysis of processes during smelting is necessary.
- You should add thermodynamic calculations. For example, FactSAGE or Thermocalc software can be used.
- Do metal samples shown in table 4 sustainable for steelmaking? How can they be used?
Some comments are placed below:
Line 96-98: The pyrometallurgical process named InduRed allows also extracted of Li
https://doi.org/10.3390/met11111844
Line 132-136: You should indicate the used type of NMC battery like NMC532, NMC622 or others.
Line 132-138: Please add the magnetic separation process parameters. What is separator type did you use? How did you receive a different type of BM? It would be best if you described it in detail.
Line 138-144: Please, add mark, brand and model for all equipment. What is method for determining chemical composition did you use?
Line 149: You should indicate the total composition of BM. What are others?
Line 196-196: Please, add the mark, brand and model of the induction furnace
Line 202-204: It is necessary to move the text
Line 205: It would be better to change the text on ‘Experimental procedure of smelting’ or similar
Line 215: You should indicate the type of thermocouple. Please, add smelting conditions for all mixtures, such as temperature and smelting duration.
Line 455: Remove figure seven. Why did you not use the flotation method to recover graphite before smelting? Is it the promising way? What do you think about it?
Author Response
The authors express their appreciation to the reviewer for the valuable comments and suggestions to improve the quality of our manuscript. Below we have provided point by point response to the reviewer comments.
The manuscript has interesting experimental results, but thermodynamic analysis of processes during smelting is necessary.
Authors’ response: The authors are grateful for the reviewer feedback and the kind revision of the manuscript.
- You should add thermodynamic calculations. For example, FactSAGE or Thermocalc software can be used.
Authors’ response: The authors appreciate the given suggestion from the reviewer. Our ongoing research aims to improve the metals recovery and enrich the Li evaporation in gas phase and collection in the dust using thermodynamic calculations such as FactSage and HSC Chemistry programmes. The thermodynamic calculations will support us to design the slag and adjust the chemistry of the black mass. The authors have considered this comment and added these clarifications in the last paragraph before the conclusions section and also in the last point in the conclusions.
- Do metal samples shown in table 4 sustainable for steelmaking? How can they be used?
Authors’ response: Yes, those metals could be of interest to be used for sustainable steelmaking as it can provide cheaper and sustainable source of alloying elements particularly Co and Ni but further studies and investigations are needed for effective utilization of these metals.
- Some comments are placed below:
Line 96-98: The pyrometallurgical process named InduRed allows also extracted of Li
https://doi.org/10.3390/met11111844
Authors’ response: The authors highly appreciate this comment and both InduRed and InduMelt has been added with new references (ref. 32 & 33).
Line 132-136: You should indicate the used type of NMC battery like NMC532, NMC622 or others.
Authors’ response: The first grade was NMC532 while the second grade is mixture of different spent LIBs types with high Co contents. This has been added to sub-section 2.1.
Line 132-138: Please add the magnetic separation process parameters. What is separator type did you use? How did you receive a different type of BM? It would be best if you described it in detail.
Authors’ response: Simple magnetic separation has been applied and the authors will consider to have more details in coming work. The BM received from one battery recycler, and they prefer not to mention the company name and the authors respect their request.
Line 138-144: Please, add mark, brand and model for all equipment. What is method for determining chemical composition did you use?
Authors’ response: the equipment used in analysis is added to sub-section 2.1.
Line 149: You should indicate the total composition of BM. What are others?
Authors’ response: The other was oxygen, and it is added to the Table 1.
Line 196-196: Please, add the mark, brand and model of the induction furnace
Authors’ response: the induction furnace is ASEA model with the melting power 90 kW and frequency 2220 Hz. This description is added to the text in sun-section 2.2.3. Reference 37 is added which give full description of the furnace with schematic diagram.
Line 202-204: It is necessary to move the text
Authors’ response: The text is moved, and the bold format is removed.
Line 205: It would be better to change the text on ‘Experimental procedure of smelting’ or similar
Authors’ response: The text changed as per reviewer request.
Line 215: You should indicate the type of thermocouple. Please, add smelting conditions for all mixtures, such as temperature and smelting duration.
Authors’ response: The thermocouple type is CELOX and it is added to the text.
Line 455: Remove figure seven. Why did you not use the flotation method to recover graphite before smelting? Is it the promising way? What do you think about it?
Authors’ response: Graphite separation by flotation is very promising method to separate graphite. One publication has been recently published by our NEXT-LIB project partner and it is included in the reference list (ref. 39). Our ongoing research is investigated the feasibility of graphite separation from pre-treated black mass in pilot scale.
The authors greatly appreciate the impression of the reviewer for our article and their comments and recommendations which have been very useful to improve the quality of the article. We uploaded the revised version with all track changes highlighted in blue color and we thank you very much for your kind consideration of this resubmitted version of our manuscript.
Best regards,
Elsayed Mousa
Corresponding author
Swerim AB
Metallurgy department
Luleå, Sweden
elsayed.mousa@swerim.se
Round 2
Reviewer 2 Report
Dear authors!
The manuscript was significantly improved after revision. So, in my opinion, it can be published in its present form.